# Quadruple-Rule-Out Computed Tomography Angiography (QRO-CT): A Novel Dual-Energy Computed Tomography Technique for the Diagnostic Work-Up of Acute Chest Pain

**DOI:** 10.3390/diagnostics13172799

**Published:** 2023-08-29

**Authors:** Turkhun Cetin, Mecit Kantarci, Baris Irgul, Sonay Aydin, Fahri Aydin, Taner Koseturk, Akin Levent

**Affiliations:** 1Department of Radiology, Erzincan Binali Yidirim University, Erzincan 24100, Turkey; turkhun.cetin@erzincan.edu.tr (T.C.); sonay.aydin@erzincan.edu.tr (S.A.); alevent@erzincan.edu.tr (A.L.); 2Department of Radiology, Ataturk University, Erzurum 25100, Turkey; akkanrad@hotmail.com (M.K.); fhraydn@gmail.com (F.A.); 3Department of Anatomy, Erzincan Binali Yidirim University, Erzincan 24100, Turkey; tanerkoseturk@gmail.com

**Keywords:** DECT, triple rule out, myocarditis, perfusion deficit, emergency

## Abstract

Background: Computerized tomography (CT) has been increasingly utilized in the differential diagnosis of acute chest pain. Combining the triple rule out CT angiography (TRO-CT) approach with dual-energy CT (DECT) can enhance the diagnostic capability by identifying myocardial perfusion deficiencies. This combination can yield a quadruple-rule-out computed tomography angiography (QRO-CT) technique. The aim of this study is to determine the efficacy of the QRO-CT. Methods: Intraluminal diseases and abnormalities in the main coronary arteries and branches were investigated. The myocardial dark spots on the color-coded iodine map were identified as perfusion deficiencies. Pulmonary arteries and aorta were also evaluated. Results: The study population consisted of 211 patients. The sensitivity, specificity, and positive and negative predictive values of QRO-CT for pulmonary embolism were 93.5%, 100%, 100%, and 95.3%, respectively. For obstructive coronary artery disease, the values were 96.1%, 93.4%, 89.2%, and 97.7%, respectively. For myocarditis, the values were 69.2%, 100%, 100%, and 93.6%, respectively. Conclusions: the QRO-CT method may successfully evaluate myocardial perfusion deficits, hence expanding the differential diagnosis capabilities of the standard TRO-CT method for myocarditis. It can provide useful information on myocardial perfusion, which may influence the choice to perform invasive catheterization in cases of coronary artery obstruction.

## 1. Introduction

Acute chest pain is a common reason for emergency department (ED) visits. Electrocardiographic (ECG) abnormalities, increased cardiac laboratory biomarkers, and characteristic symptoms can all be used to diagnose some patients. Normal ECG results or normal cardiac biomarkers, on the other hand, do not exclude acute cardiovascular illness, and symptoms may be atypical. Imaging methods, especially computerized tomography (CT), have been increasingly utilized in the differential diagnosis of acute chest pain [1,2]. 

Triple rule out computed tomography angiography (TRO-CT) is a customized ECG-gated examination which can assess the aorta, the coronary arteries, pulmonary arterial circulation, and the middle–lower parts of the lungs in the same imaging session. It is a useful technique for detecting coronary artery disease (CAD), pulmonary embolism (PTE), and acute aortic syndromes (AAS) with increased z-axis coverage and, as a result, improved temporal resolution and lower radiation exposure. It is previously stated that, in about 75% of patients, TRO-CT can safely avoid the need for performing additional diagnostic tests [3,4].

Dual-Energy Computed Tomography (DECT) is a developing technology that provides information about the material composition via image acquisition by varying photon energy levels [5]. In the last decade, DECT has been increasingly utilized for cardiac imaging [6,7]. Different energy levels of X-ray are used to penetrate an iodinated contrast medium, and it creates unique absorption characteristics. As a result, iodine mapping reveals the distribution of iodine in the myocardium [6], where the dark areas indicate a lack of iodine. DECT detects cardiac perfusion defects very precisely [8,9].

DECT can be used to evaluate the morphology and functionality of the myocardial blood supply. Using single photon emission computed tomography and invasive coronary angiography as reference standards, DECT successfully identified coronary artery stenosis and myocardial ischemia. DECT can precisely deliver integrated imaging of myocardial perfusion and coronary artery morphology during the same scan [10].

Recent investigations have shown that LGE-MRI and late iodine enhancement (LIE)-DECT can both detect chronic MI. In addition to the excellent diagnostic performance of the 80 kVp and 135 kVp image series, linear blending of low/high voltage LIE-DECT images resulted in a similar but slightly improved diagnostic performance for cardiac scar tissue. Additionally, the measurements of the infarct volume in the 80 kVp image series and the linear blending image series had the best agreement with the measurements of the infarct volume in the late gadolinium enhancement-MRI. Iodine distribution maps, however, demonstrated limited sensitivity (52% on the segment-based analysis and 34% on the segment-based analysis) in terms of detecting myocardial LIE [11].

Detecting myocardial perfusion deficiencies is critical for a variety of reasons. Acute myocarditis can cause chest pain and myocardial perfusion abnormalities. Additionally, the extent and severity of myocardial perfusion deficits are critical criteria for determining the necessity and efficacy of cardiac catheterization in patients with acute coronary syndrome [12,13].

Combining the TRO-CT approach with DECT technology in cases of acute chest pain can enhance diagnostic capability beyond the differentiation of CAD, PTE, and AAS. By identifying myocardial perfusion deficiencies, this combination can yield a quadruple-rule-out computed tomography angiography (QRO-CT) technique. The purpose of this study is to determine the efficacy of the QRO-CT in diagnosing and managing acute chest pain.

## 2. Materials and Methods

The local institutional review board approved this prospective study, and informed consent was acquired from all the participants. The study was conducted between January 2021 and December 2021. 

### 2.1. Patients

The current study included all of the patients who applied to the ED with acute chest pain and were directed to the radiology department for CT imaging. The patients who were not eligible for TRO-CT study were excluded. Patient selection criteria for TRO-CT were acquired from Halpern’s study (Table 1) [3]. Patients with a contraindication for the contrast medium use were also excluded. Accordingly, 269 patients were included; however 58 patients were excluded. The study population consisted of 211 patients.

Five patients were excluded from the study because of contraindications to the use of the contrast medium. Also excluded from the study were 15 patients with a history of bypass or stenting, 13 patients who did not have adequate kidney functions for the use of contrast media, 11 patients with cardiac arrhythmias, and 14 patients with abnormal ECG findings.

Final diagnoses of the patients were defined via clinical work-up and follow-up data. 

### 2.2. DECT Protocol

The DECT images were created via a 64-slice dual-source multi-detector CT scanner (Somatom Definition Flash, Siemens Healthcare, Forchheim, Germany). The protocol begins with a prospectively ECG-triggered CTA acquisition between the levels of the carina and the base of the diaphragm. According to patient’s condition, ECG-triggered (prospective), ECG-gated (retrospective), or prospectively ECG-triggered spiral acquisition was used with following parameters: 80/140 Sn kVp, 60 mAs, rotation time 0.28 s, slice thickness 1.5 mm, and pitch value: 3.0. An intravenous iodinated contrast material of 60 to 90 mL was administered at an injection rate of 4 to 6 mL/s, followed by a saline chaser of 50 mL. Nitroglycerin or beta-blocker administration was not used. First, the coronary artery opacification occurs; meanwhile, homogeneous enhancement of the pulmonary arteries happened. MIP, CPR, and volume-rendered reconstructions were obtained in order to evaluate the aorta, coronary, and pulmonary arteries; axial CT images were used to confirm the findings. The CT dose index volume and the dosage–length product of the scans were noted. Patients were encouraged to adopt the deep-inspiration breath-hold technique during the procedure. The reconstruction window of the initial axial pictures was set at 75% and 45% for the cardiac cycle (end of systolic phase).

For the myocardial evaluation, the high- and low-voltage information was reconstructed via a dual-energy convolution core (D30f) that had a temporal resolution of 140 milliseconds and a thickness of 1.5 mm, with 1 mm increments utilized to maximize the signal–noise ratio. Following that, the final data were analyzed using a three-material decomposition software platform (Syngo Multimodality Workplace; Siemens, Erlangen, Germany).

### 2.3. Image Analysis

Two blinded radiologists (18 and 6 years of experience in cardiac imaging) independently examined the CT images using the 17-segment model according to the American Heart Association classification of the segmentation of the left ventricular myocardium. Additionally, intraluminal diseases and abnormalities in the main coronary arteries and branches were investigated. For the purpose of assessing CAD, any stenosis larger than or equal to 50% was considered obstructive. All main coronary arteries (left main, anterior descending, and circumflex arteries, and right coronary artery) were examined for stenosis using quantitative measurements, as were all important obtuse marginal and diagonal branches. Pulmonary arteries and the aorta were also evaluated. 

Prior to analyzing the myocardium with DECT, the ‘DE normalize contrast’ process was used on the workstation to provide consistency for the visual evaluation and to eliminate any bias due to interobserver variability. Myocardial assessment was performed using arterial phase pictures. The dark spots on the color-coded iodine map were identified as perfusion deficiencies for each subject and section. Following segment-by-segment analysis, the perfusion deficiencies on the iodine map pictures on DECT were identified. The anatomic location (transmural, intramyocardial, subepicardial, subendocardial) of the perfusion deficits was also noted. 

### 2.4. Assessment of Image Quality

Individual evaluations were conducted by two radiologists with 18 years and 6 years of experience in cardiovascular imaging. On a 5-point Likert scale, both readers rated images for overall image quality, noise, artifacts, and sharpness (5 was the best outcome and 1 was the worst). For each pathology, quality assessments were conducted independently (coronary artery disease, pulmonary embolism, myocardial perfusion deficits, and acute aortic syndromes). As final data, the mean of the quality rates was employed.

### 2.5. Statistical Analysis

SPSS version 20.0 was utilized (SPSS Inc., Chicago, IL, USA). To evaluate the normally distributed data, the Kolmogorov–Smirnov test was utilized. Numerical variables with a normal distribution were expressed as mean ± standard deviation. Number (*n*) and percentage values were used to denote categorical variables (%). Sensitivity, specificity, and positive and negative predictive values (with a 95% confidence interval (CI)) were utilized to assess the diagnostic accuracy of QRO-CT in patients with pulmonary embolism, coronary artery disease, aortic dissection, and the presence of myocardial perfusion deficits. Observers 1 and 2 independently assessed the CT images. Cohen’s Kappa coefficient was used to determine whether Observer 1 and Observer 2 were in agreement for pathology presence and quality assessment. Accordingly, the degree of agreement was classified as slight if the coefficient was between 0 and 0.20, fair if the coefficient was between 0.21 and 0.40, moderate if the coefficient was between 0.41 and 0.60, substantial if the coefficient was between 0.61 and 0.80, and almost perfect if the coefficient was between 0.81 and 1.00 [14]. A value of *p* < 0.05 was accepted as statistically significant.

Our study was approved by the ethics committee. The ethics committee approval number is as follows: 14.12.2021-EBYU-KAEK-21/12-18. 

All procedures performed in studies involving human participants were in accordance with the ethical standards of the institutional and/or national research committee and with the 1964 Helsinki declaration and its later amendments, or comparable ethical standards.

## 3. Results

The study population consisted of 211 patients. The mean age of the population was 66.13 ± 9.12 years (min–max, 45–83 years). A total of 125 (59.2%) of the patients were male and 86 (40.7%) of them were female. 

A total of 72 patients (34.1%) had a confirmed pathology. Detailed numbers of final diagnoses can be seen in Table 2. Table 3 details the image quality assessment for each pathology and the interobserver variability data for quality assessments. The mean radiation dose parameters for the DECT scans was dose–length product, 315.1 ± 23.51 mSv × cm; volume CT dose index, 7.05 ± 1.82 mSv.

Two cases of distal pulmonary embolism could not be diagnosed with CT scan, and no false positive results were detected (Figure 1). Sensitivity, specificity, and positive and negative predictive values of quadruple-rule-out CT for pulmonary embolism were 93.5% (95% confidence interval [CI] 90–100%), 100% (95%CI 96–100%), 100% (95%CI 97–99%), and 95.3% (95%CI 93–100%), respectively. Cohen’s coefficient of variation (ĸ value) for the presence of pulmonary embolism was 0.95 (almost perfect).

The results of invasive angiography verified the presence of coronary artery diseases (CAD).

Three cases were overestimated as having obstructive coronary artery disease, and one case was underestimated as having negative obstructive coronary artery disease; all of the erroneous results were caused by calcified aterosclerotic plaques. Sensitivity, specificity, and positive and negative predictive values of quadruple-rule-out CT for obstructive coronary artery disease were 96.1% (95% confidence interval [CI] 93–99%), 93.4% (95%CI 90–99%), 89.2% (95%CI 86–94%), and 97.7% (95%CI 94–100%), respectively (Figure 2). Cohen’s coefficient of variation (ĸ value) for the presence of obstructive CAD was 0.78 (substantial).

In 13 cases, myocardial perfusion deficits (PD) were identified (13/211, 18%). Cardiac magnetic resonance imaging and myocardial perfusion scintigraphy confirmed the presence of these myocardial PDs. Four of them were associated with substantial obstructive coronary artery disease. Myocardial perfusion abnormalities were observed to be associated with myocarditis in 9 individuals; however, finally, 13 patients were diagnosed with myocarditis (5 of them were identified with COVID-19-associated myocarditis, while the remaining 8 were diagnosed with nonspecific viral myocarditis). We could not define any myocardial PD in four of the myocarditis cases. Sensitivity, specificity, and positive and negative predictive values of quadruple-rule-out CT for myocarditis were 69.2% (95% confidence interval [CI] 65–74%), 100% (95%CI 96–100%), 100% (95%CI 96–100%), and 93.6% (95%CI 90–97%), respectively. Cohen’s coefficient of variation (ĸ value) for the presence of myocardial PD was 0.73 (substantial).

Perfusion deficits were found on DECT iodine map images with a subendocardial (*n* = 4, 30.7%), subepicardial (*n* = 7, 53.8%), and transmural (*n* = 2, 15.3%) anatomical location within the left ventricular wall (Figure 3). Subendocardial involvements were found in the patients who had obstructive CAD, and PDs correspond to associated coronary artery territories. Subepicardial and transmural PDs did not correspond to any coronary artery territories. ROI was used to measure the perfusion deficit and normal areas on the iodine map images on DECT. The mean value of the perfusion deficit areas was 162.13 ± 21.31 HU (range, 45–294), while that of the normal areas was 50.11 ± 9.32 HU (range, 23–91).

All of the aortic dissection cases were successfully detected with quadruple-rule-out CT angiography protocol (Figure 4). Sensitivity, specificity, and positive and negative predictive value of quadruple-rule-out CT for aortic dissection were 100% (95% CI). Cohen’s coefficient of variation (ĸ value) for the presence of aortic dissection was 1 (perfect).

## 4. Discussion

The acquisition of two distinct datasets using two different X-ray spectra energies—”low kilovoltage peak (kVp)” and “high kVp” spectra—makes up the new imaging technique known as dual-energy computed tomography (DECT).

Any given substance exhibits various X-ray absorption properties at various energies. Because of these variations, DECT offers many more uses than conventional single-energy CT (SECT), which only collects anatomical and density data.

Since its introduction, a number of cardiac DECT applications have been investigated, including virtual monoenergetic images (VMI), virtual non-contrast (VNC) reconstructions, and iodine myocardial perfusion maps. These applications have been shown to increase diagnostic accuracy and image quality while decreasing the administration of radiation and contrast media.

Although DECT was first introduced more than ten years ago, it has taken longer than expected for this technology to become widely used. This is largely because there aren’t enough scanners available, there isn’t widespread clinical validation, and there aren’t enough prognostic studies that show DECT is better than traditional imaging methods. Several studies are currently being conducted to enhance the DECT’s image quality and diagnostic precision, leading to the growth of cardiac CT applications [15].

Myocardial infarction (MI) and coronary artery disease are among the leading causes of death in western societies. It has been demonstrated that, while deciding whether to continue treatment for patients with acute coronary syndrome, the distinction between viable and nonviable myocardium is critical. Myocardial segments that are akinetic but still alive (hibernating) profit greatly from interventional myocardial revascularization. Myocardial revascularization, on the other hand, does not benefit akinetic or nonviable myocardial segments, and the hazards of the surgery are still there.

Myocardial late enhancement (LE) is a condition that can occur in both acute and chronic MI and is thought to represent evidence of nonviable myocardium. Since the LE magnetic resonance imaging (LE-MRI) technique has previously demonstrated its ability to distinguish between viable and nonviable myocardium and acute and chronic MI, it is currently a trusted tool for routinely examining patients with coronary artery disease [16].

A recent study found that 10 min after the injection of the contrast agent is the ideal time to look for myocardial LE using LE-DECT. Similar results for single-source MDCT have already been published. LE-MRI exhibits greater sensitivity, specificity, PPV, and accuracy when compared to LE-DECT 10 min after the injection of the contrast agent (0.79 vs. 0.77, 0.94 vs. 0.92, 0.76 vs. 0.69, 0.94 vs. 0.94, and 0.9 vs. 0.89, respectively). Although these differences are not statistically significant, they might suggest that, even when dual-energy CT is used, LE-MRI is still preferable to LE-CT imaging. Although there was no discernible difference in specificity and diagnostic accuracy between LE-DECT and LE 100 kV CT images, LE-DECT had consistently greater sensitivities for identifying MI [16].

DECT has been proven to be beneficial in viral myocardiopathies in addition to ischemic instances. In a recent study, our research team compared the COVID-19 positive case group to the control group, which consisted of patients with a clinical suspicion of coronary artery disease, utilizing the dual-source CT angiography technique. Between the groups, there was no discernible difference in the severity of coronary luminal anomalies. However, the COVID-19 case group exhibited significantly greater myocardial perfusion anomalies than the controls. Additionally, 100% agreement between the two observers revealed perfusion deficits. These regions of perfusion deficit, which correspond to the dark lesions in the iodine mapping approach, were determined to have significant myocardial damage [17].

Although EMB is the gold standard for diagnosis of acute myocarditis, non-invasive methods have gained popularity due to EMB’s invasiveness and lack of sensitivity. As a result, T2-weighted imaging and LGE in CMR have gained in popularity. Previous research has demonstrated a histological correlation between CMR and areas of acute inflammation.

The delayed enhancement regions were shown in certain investigations using cardiac CT to be comparable to those detected on CMR in acute myocarditis. With a single contrast-enhanced CT image taken in a matter of seconds utilizing a small radiation dose, the DECT technique is utilized in cardiac imaging to reveal details on the coronary artery system and myocardial perfusion. Similar to single-energy CT, DECT scans can be acquired with the proper dosages. Our research team’s prior investigation on cardiac DECT came to the conclusion that CMR in cases of acute myocarditis was closely connected with the dark regions on the color-coded iodine map on DECT. Acute myocarditis, coronary artery disease, and malignant coronary artery abnormalities, which can result in symptoms of acute coronary syndrome, can all be accurately identified by combining CT coronary angiography and an iodine map with a DECT in a single examination [18].

The purpose of this study is to define and demonstrate a novel diagnostic approach, quadruple-rule-out computed tomography angiography (QRO-CT), for evaluating acute chest pain in the emergency scenario. In addition to the previously established indications for TRO-CT, the QRO-CT approach can identify myocardial perfusion deficits and provide diagnostic hints for myocardial ischemia/infarction or myocarditis cases by combining TRO-CT with DECT technology.

TRO-CT is an effective protocol for the diagnostic work-up of acute chest pain. It is a useful method for the exclusion and the diagnosis of CAD, PTE, and AAS with high specificity; hence, it can play an important role in decision-making in the ED. TRO-CT can define CAD and PTE with a satisfying sensitivity and specificity in comparison with dedicated protocols [19,20]. We have used basically a similar protocol with TRO-CT, with the addition of DECT, to assess myocardial perfusion deficits. A key point for an effective TRO-CT is appropriate patient selection. A high negative predictive value, an advantageous aspect in ED imaging, is contingent upon optimal patient selection, allowing for the reliable and expeditious discharge of patients after a negative CT scan [3,20]. TRO-CT is also a cost-effective approach due to its high negative predictive value, which is obtained by careful patient selection [3]. We have applied the same patient selection criteria with the TRO-CT protocol in our study population.

As expected, by using a similar protocol and patient selection criteria, we have obtained similar diagnostic accuracy values with previous TRO-CT studies in PTE, CAD, and aortic dissection groups [19,20,21]. Similarly with TRO-CT studies, QRO-CT protocol managed to diagnose PTE with a sensitivity and specificity of 93.5% and 100%, respectively; obstructive CAD with 96.1% and 93.4%, respectively; and aortic dissection with 100%.

Various dosage estimates have been published for the TRO-CT protocol. The mean dosage of radiation varies between 8.6 and 19.4 mSv in several studies [4,21,22,23]. A benefit of DECT systems is that they can reduce the radiation dose by increasing the pitch value. We used CT scans with a pitch of 3.0 and a mean radiation dosage of 315.1 ± 23.51 mSv × cm, 7.05 ± 1.82 mSv, which is less than previous studies. We have found only one study with a better dose exposure using the TRO-CT method (5.7 ± 2.7 mSv), which used sinogram affirmed iterative reconstruction to reduce radiation doses by 34% [4].

Even though most major causes of acute chest pain may be evaluated and identified safely by TRO-CT, the actual list of differential diagnosis is bigger and involves pathologies which cannot be detected with the TRO-CT method, such as myocarditis and pericarditis [24]. Along with the advances in DECT technology, the diagnostic capacity of CT has been enlarging, especially in the emergency setting. DECT is used in cardiac imaging to provide information about the coronary artery system and myocardial perfusion by using a single dose of contrast medium obtained within a short time and with a relatively lower radiation dose [8,12]. A case study of two patients with acute myocarditis showed focal myocardial PD areas corresponding to high signals on the T2-weighted cardiac magnetic resonance (CMR images) and late Gadolinium enhancement (LGE), [25] which were attributed to edema. In another case report, it was found that abnormally delayed iodine enhancement areas on DECT showed an excellent topographic match with CMR [26]. Detecting PDs is also critical for assessing myocardial viability and catheterization indications [12].

The QRO-CT method successfully detected all of the PDs associated with obstructive CAD in the current study. Additionally, it may identify a considerable number of PDs caused by myocarditis. We have shown that the QRO-CT method can diagnose myocarditis cases with a sensitivity, specificity, and positive and negative predictive value of 69.2%, 100% (95% CI 96–100%), 100% (95% CI 96–100%), and 93.6%, respectively. QRO-CT can detect myocarditis cases with a better specificity and positive predictive value then its sensitivity and negative predictive value, indicating that, even though QRO-CT might miss some of the myocarditis cases, a positive finding for myocarditis can be accepted as a reliable indicator. Also, our myocarditis case sample has outnumbered previous similar studies [25,26].

The current study has certain limitations that are worth noting. A larger patient population may result in more trustworthy results. Patients were recruited using the TRO-CT protocol criteria; new and more focused recruitment criteria for the QRO-CT method may improve diagnostic accuracy. Further research is needed to provide more appropriate selection criteria and accurate results. We did not apply any premedication before the QRO-CT scan; as a result, an insufficient heart rate may generate explicit artifacts, which may limit the technique’s wider use in the emergency setting. Additional research is needed to identify the effect of premedication. Very few of the patients included in the study had troponin values taken during the shooting. For this reason, these data could not be included in the study. The diagnoses of the cases included in the study were confirmed by clinical follow-ups. Since MRI was not available for all patients, the investigated method could not be compared with a more known method, MRI. This subject can be considered as an inspiration for future studies.

Finally, the evaluation was partially unsatisfactory in locations other than the left ventricular free wall due to beam-hardening artifacts.

## 5. Conclusions

According to our first and preliminary result, the QRO-CT method—a hybrid of TRO-CT and DECT myocardial perfusion evaluation—can successfully evaluate myocardial perfusion deficits, hence expanding the differential diagnosis capabilities of the standard TRO-CT method for myocarditis. Additionally, it can provide useful information on myocardial perfusion, which may influence the choice to perform invasive catheterization in cases of coronary artery obstruction.

Prospective studies to be carried out in the future will both confirm our hypotheses and provide an opportunity for the technique we have introduced to become widespread.

## Figures and Tables

**Figure 1 diagnostics-13-02799-f001:**
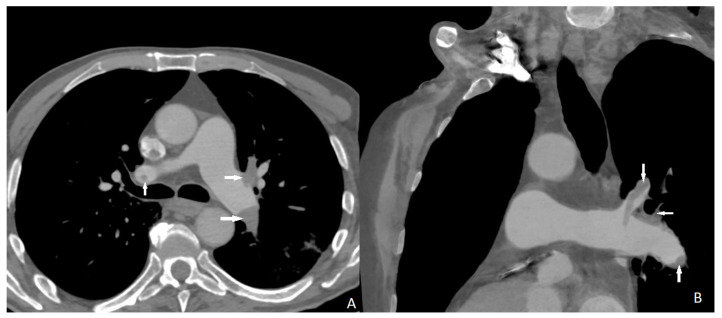
Filling defects (arrows) on pulmonary arterial phase in the both main and left upper lobar pulmonary artery consistent with pulmonary embolus. ((**A**) axial; (**B**) coronal).

**Figure 2 diagnostics-13-02799-f002:**
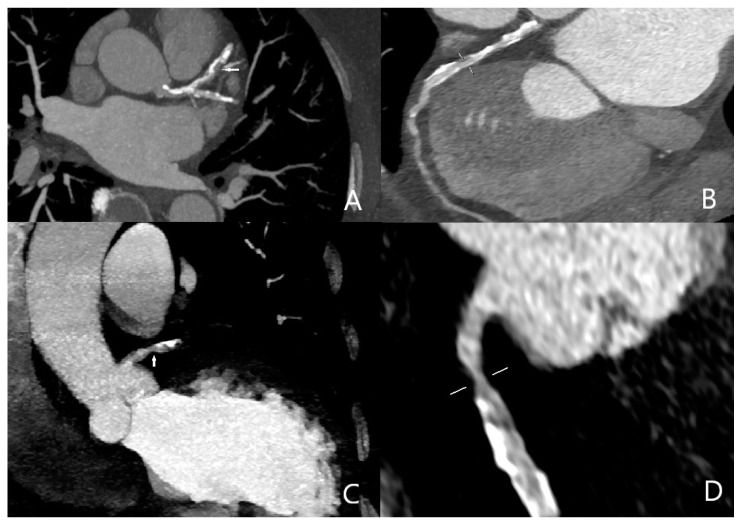
Obstructive coronary artery disease in the midportion of the left anterior descending artery ((**A**) arrow, (**B**) marking) and in the proximal portion of the right coronary artery ((**C**) arrow, (**D**) marking).

**Figure 3 diagnostics-13-02799-f003:**
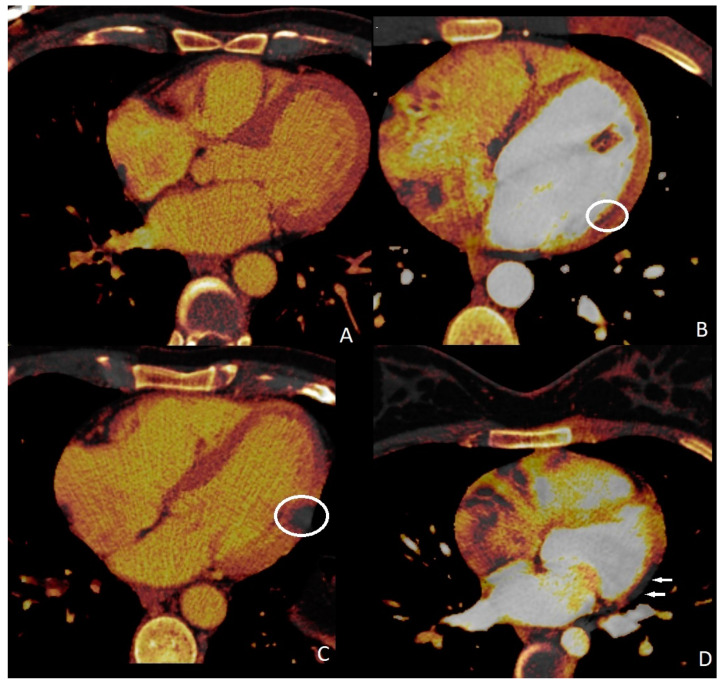
Myocardial perfusion maps. Normal (**A**) perfusion image, subendocardial ((**B**), circle), transmural ((**C**), circle), and subepicardial ((**D**), arrows) perfusion deficits.

**Figure 4 diagnostics-13-02799-f004:**
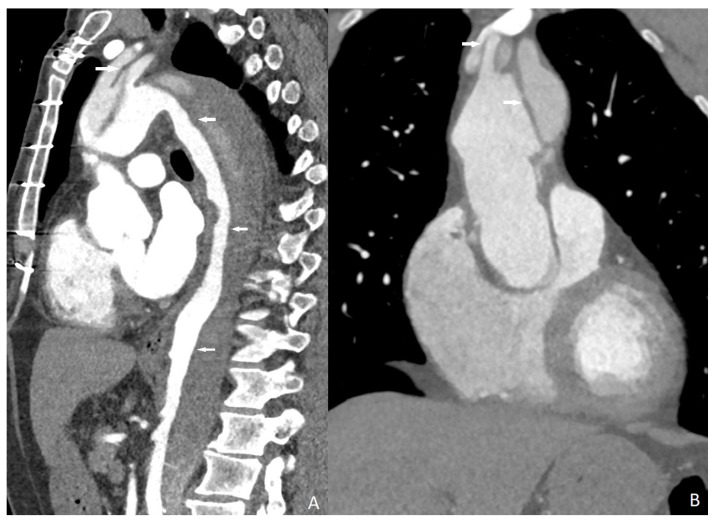
Stanford type B aortic dissection affecting both ascending and descending aorta (arrows). ((**A**) sagittal; (**B**) coronal).

**Table 1 diagnostics-13-02799-t001:** Patient selection criteria.

Low to Moderate Clinical Risk for ACS
Clinically non-ACS diagnosis is considered
Normal ECG or nonspecific changes
Absence of extensive coronary calcium presence risk
Absence of bypass or stent history
Ability to tolerate CT and hold breath
Cardiac rhythm acceptable for a ECG gated scan
Adequate renal functions for iodine based contrast medium injection
ACS: Acute coronary syndrome ECG: Electro cardiogram CT: computed tomography

**Table 2 diagnostics-13-02799-t002:** Final diagnosis.

Diagnosis	Number (%)
Pulmonary Embolism	31 (43%)
Obstructive Coronary Artery Disease	26 (36.1%)
Myocarditis	13 (18%)
Aortic Dissection	5 (6.9%)
Total	72 (100%)

**Table 3 diagnostics-13-02799-t003:** Image quality assessments for each subgroup and Cohen’s ĸ value for interobserver variability.

Quality Score	1 *n* (%)	2 *n* (%)	3 *n* (%)	4 *n* (%)	5 *n* (%)	Total *n* (%)	Cohen’s ĸ Value
Pulmonary Embolism	0	0	14 (6.6%)	43 (20.3%)	152 (72%)	211 (100%)	0.87 (almost perfect)
Obstructive Coronary Artery Disease	8 (3.7%)	15 (7.1%)	17 (8%)	63 (29.8%)	108 (51.1%)	211 (100%)	0.83 (almost perfect)
Myocardial perfusion deficits	19 (9%)	9 (4.2%)	16 (7.5%)	80 (37.9%)	87 (41.2%)	211 (100%)	0.82 (almost perfect)
Aortic Dissection	0	0	0	41 (19.4%)	170 (80.5%)	211 (100%)	0.91 (almost perfect)

## Data Availability

The data that support the findings of this study are available from the corresponding author (BI) upon reasonable request.

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
