# Peer review of "Quadruple-Rule-Out Computed Tomography Angiography (QRO-CT): A Novel Dual-Energy Computed Tomography Technique for the Diagnostic Work-Up of Acute Chest Pain"

_diagnostics, 2023, doi:10.3390/diagnostics13172799_

Round 1
Reviewer 1 Report
Throughout the text please correct spacing.
Line 86 - did you wait for the serum troponin levels before doing the CT scan? If yes please provide the number/percentage of patients scanned with positive troponin levels
Line 88 - please provide reasons why 58 patients were excluded.
Line 168 - did you mean: "Two cases of distal pulmonary embolism could not be diagnosed with CT scan...?"
Line 224 - please correct "4" to "for"
The English Language requires only minor revisions.
Author Response
Response
Line 86 - Very few of the patients included in the study have troponin values ​​taken during the shooting. For this reason, this data could not be included in the study. We have added these sentences to the limitation.
Line 88- 5 patients were excluded from the study because of contraindications to the use of contrast medium. They were excluded from the study because 15 patients had a history of bypass or stenting, 13 patients did not have adequate kidney functions for the use of contrast media, 11 patients had cardiac arrhythmias and 14 patients had abnormal ECG findings. We have added these sentences to the relevant section.
Line 168- We apologize for the mistake. Yes, we meant it. "Two cases of distal pulmonary embolism could not be diagnosed with CT scan...?"
Line 224- We apologize for the mistake. Changed "4" to "for".

Reviewer 2 Report
The authors present a quadruple-rule-out technique to rule out coronary artery disease, acute aortic syndrome, pulmonary embolism and myocarditis in patients that acutely present with chest pain. They found very good predictive value of the new technique that uses dual energy CT on top of traditional triple-rule-out CT.
While this technique seems promising, I don't know if the study has been conducted accordingly to allow the exact calculation of predictive value, as I did not find a clear comparison with the gold standard (for myocarditis, MRI for example). Therefore, it should be seen as hypothesis generating idea, with no recommendation of use elsewhere. Furthermore, the data a quite old.
Statistics and references seem adequate.
Author Response
Response
The diagnoses of the cases included in the study were confirmed by clinical follow-ups. Since MRI was not available for all patients, the investigated method could not be compared with a more known method, MRI. This subject can be considered as an inspiration for future studies.
Prospective studies to be carried out in the future will both confirm our hypotheses and provide an opportunity for the technique we have introduced to become widespread.
We have added these sentences to the limitation and conclusions.

Round 2
Reviewer 2 Report
The authors have assessed all my comments accordingly.